# Surgical Correction of Infundibular Muscular Ventricular Septal Defect in a Cat: A Case Report

**DOI:** 10.3390/ani14121736

**Published:** 2024-06-08

**Authors:** Takuma Aoki, Tomomi Terakado, Yao Jingya, Kentaro Iwasaki, Hayato Shimoda, Naoyuki Fukamachi, Takashi Miyamoto

**Affiliations:** 1Laboratory of Small Animal Surgery, Department of Veterinary Medicine, School of Veterinary Medicine, Azabu University, Sagamihara City 252-5201, Kanagawa, Japan; kj2027@azabu-u.ac.jp (T.T.); dv2103@azabu-u.ac.jp (Y.J.); k-iwasaki@azabu-u.ac.jp (K.I.); 2Azabu University Veterinary Teaching Hospital, Azabu University, Sagamihara City 252-5201, Kanagawa, Japan; 3Gunma Children’s Medical Center, Shibukawa City 377-8577, Gunma, Japan; h-shimoda@gcmc.pref.gunma.jp (H.S.); n.fukamachi7786@gmail.com (N.F.); 4Kodama Kyodo Hospital, Setagaya Ward, Tokyo 156-0052, Japan; guuji3838@gmail.com

**Keywords:** cat, congenital heart disease, feline cardiac surgery, surgical correction, ventricular septal defect

## Abstract

**Simple Summary:**

In a groundbreaking procedure, our team successfully performed surgery on a young Ragdoll cat with a rare type of heart defect known as an infundibular muscular ventricular septal defect, which is a hole between the two lower heart chambers. This condition, if left untreated, can lead to severe heart problems. Therefore, the surgery was performed using a method called cardiopulmonary bypass, which is quite unusual in cats. This approach involved stopping the heart temporarily to repair the defect safely. Despite being a challenging situation, in which an adjacent structure was unintentionally damaged intraoperatively, resulting in a temporary complication that was treated non-surgically, the cat recovered well. By 490 days postoperatively, the cat was healthy with only a minor residual defect, showing that the heart had nearly returned to its normal size. This case highlights a promising surgical option for treating similar heart defects in cats, which can potentially improve their quality of life significantly. Therefore, with proper surgical treatment, even in cases of severe congenital heart disease in cats, owners will be able to live with their cats without the need for numerous medications.

**Abstract:**

Ventricular septal defects (VSDs) can lead to congestive heart failure and pulmonary hypertension, particularly in patients with large shunts. However, no surgical treatment for feline VSD has been reported. Here, we elucidated the first surgical correction of an infundibular muscular VSD in a one-year-old Ragdoll cat, atypically located and classified under the Soto classification rather than the standard Kirklin classification, through cardiac arrest using cardiopulmonary bypass—a method rarely used in feline cardiac surgery. Detailed echocardiography revealed that the defect required intervention owing to left heart and main pulmonary artery enlargement. Despite the VSD being located on the contralateral side, as anticipated in the preoperative examinations, the choice of median sternotomy allowed for the successful closure of the defect. Conversely, the insertion of two cannulas into the ascending aorta resulted in damage to the adjacent thoracic duct, causing transient chylothorax, which was resolved with conservative treatment. Cardiac arrest induced by a cardioplegic solution facilitated the surgical procedure, although it leads to anemia in cats. However, on postoperative day 490, the patient exhibited only minor residual shunting, with normalized heart size, and remained healthy. This technique appears to be a viable treatment option for congenital heart disease in cats.

## 1. Introduction

Ventricular septal defect (VSD) is a congenital heart disease characterized by an unclosed ventricular septum leading to left-to-right shunting, which can result in congestive left heart failure and pulmonary hypertension in patients with large defects and significant shunt volumes. The prevalence of congenital heart disease in mixed-breed cats is 0.14%, indicating its rarity in cats [1]. However, among these congenital heart diseases, VSD accounts for 21%. Other clinical studies have found that 8% of cats with heart disease have congenital heart disease, and 50% of these cases are due to VSD, making it the most common congenital heart disease in cats [2]. In cats, VSD predominantly presents as the membranous or perimembranous type, accounting for approximately 79% of cases. Other types include the supracristal type at approximately 13% and the atrioventricular canal and muscular types at approximately 4% of cases each [3]. These classifications are typically referred to as the Kirklin classification and are widely used in veterinary medicine [4]. VSD alone rarely presents any clinical signs; however, among 27 cats with VSD, 6 exhibited respiratory symptoms, 3 of whom developed congestive left heart failure [3]. Typically, medical management such as diuretic therapy is implemented; however, palliative surgical treatments such as pulmonary artery banding to restrict blood flow have also been reported [5]. 

In humans, cases of VSD with pulmonary hypertension or other clinical symptoms are treated with catheter intervention or direct closure under cardiopulmonary bypass during cardiac arrest [6]. To our knowledge, no surgical treatment for feline VSD has been reported. Therefore, this report presents a case that did not fall under the Kirklin classification but was diagnosed as an infundibular muscular VSD according to the Soto classification [7], and underwent surgical correction using cardiopulmonary bypass.

## 2. Case Description

### 2.1. Patient

A three-month-old female Ragdoll cat weighing 1.98 kg (body condition score [BCS]: 4/9) was presented for consultation. No apparent cardiogenic clinical signs were observed; however, a heart murmur was observed before acquisition. During vaccination, chest radiography revealed cardiac enlargement at a referral hospital based on subjective evaluation, prompting further examination at the Azabu University Veterinary Teaching Hospital. The heart murmur was most pronounced at the right cardiac apex, with an audible grade 5/6 systolic murmur. The body temperature was 38.2 °C, the heart rate was 138 bpm, and the respiratory rate was 48 breaths/min. Chest radiography indicated a vertebral heart size of 9.0 vertebrae (v), which is within the normal range for a kitten (median 9.5 v; range: 7.7–10.8 v) [8]. The cardiothoracic ratio was 74.8%, which falls within the normal range for kittens (median 67.2%, range 49.2–85.0%; Figure 1). Electrocardiography revealed increased R waves, but the electrical axis was +80° with a normal sinus rhythm (180 bpm), and no arrhythmias were observed. Transthoracic echocardiography (TTE) was performed using an ultrasound unit (Vivid E9; GE Healthcare Co., Ltd., Tokyo, Japan) equipped with 6–12 MHz phased-array transducers. TTE identified a defect just below the aortic valve in the right parasternal left ventricular outflow tract view. Additionally, at the level of the aortic valve in the right parasternal short-axis view, the defect was located between the pulmonary artery outflow tract and tricuspid valve, leading to a diagnosis of a combination of Type 2 and 3 VSD according to the Kirklin classification (Figure 2A). The VSD flow was 4.97 m/s. The left atrial diameter was 1.45 cm, which was enlarged compared with the normal value for a 2 kg kitten (average, 0.96 cm; 95% prediction intervals [95% confidence interval (CI)]: 0.82–1.13 cm) [8]. The left ventricular end-diastolic diameter (LVIDd) was 1.94 cm, which was larger than the normal value of 1.25 cm (95% CI: 1.04–1.49 cm) [8].  The interventricular septum in diastole (IVSd) and left ventricular free wall in diastole (LVFWd) were 0.38 cm and 0.43 cm, respectively, values which were within the normal range for a 2 kg kitten (IVSd: 0.37 cm [95% CI: 0.29–0.48 [8]]; LVFWd: 0.36 cm [95% CI: 0.27–0.48 cm [8]]). The interventricular septum in systole (IVSs) and left ventricular free wall in systole (LVFWs) were 0.60 cm and 0.62 cm, respectively, which were within the normal range for a 2 kg kitten (IVSs: 0.59 cm [95% CI: 0.46–0.77 [8]]; LVFWs: 0.64 cm [95% CI: 0.51–0.79 cm [8]]). The left ventricular end-systolic diameter (LVIDs) was 0.90 cm, and the fractional shortening (FS%) was 53.57%. Significant enlargement of the pulmonary artery was evident (main pulmonary artery-to-aorta diameter ratio: MPA/Ao, 1.48 (0.95/0.64 cm); normal value in cats < 0.90; Figure 2B) [9]. The pulmonary artery was significantly dilated, and the pulmonary-to-systemic blood flow ratio (Qp:Qs) [10] calculated using Doppler imaging was 2.45. The owner expressed interest in surgical treatment for the patient and visited our hospital but decided to wait for the kitten to grow up due to its small size.

On day 383, the cat weighed 4.44 kg (BCS: 4/9) with no clinical signs. Echocardiography revealed that the VSD had minor and major axes of 1.2 mm and 8.1 mm, respectively (Figure 3A), with a VSD/Ao ratio of 95.3%. Color Doppler imaging in the right parasternal short-axis view at the level of the aortic valve revealed a shunt flow (5.01 m/s) at the 12 o’clock position (Figure 3B). The LA/Ao ratio was 1.69 (1.43/0.85 cm; normal range < 1.5 [11]), which was considerably enlarged, and LVIDd was 1.8 cm, which was at the upper limit of the normal range for an adult cat weighing 4.5 kg (1.58 cm, range: 1.27–1.98 cm [12]). IVSd and LVFWd were 0.40 cm and 0.53 cm, respectively; IVSd was within the normal range (median 0.39 cm, interquartile range [IQR]: 0.29–0.51 cm [12]), whereas LVFWd was mildly enlarged (0.38 cm, IQR: 0.29–0.50 cm) [12]. IVSs and LVFWs were 0.67 cm and 0.68 cm, respectively; they were within the normal range (IVSs: median 0.62 cm, IQR: 0.44–0.87 cm; LVFWs: 0.65 cm, IQR: 0.48–0.87 cm [12]). LVIDs was 0.98 cm in the normal range (median 0.86 cm, IQR: 0.57–1.30 cm), and FS% was 45.51% (normal range: median 45%, IQR: 28–62% [12]).

Significant enlargement of the pulmonary artery was evident (MPA/Ao: 1.29 (1.10/0.85 cm); normal value in cats < 0.90 [9] (Figure 4). No tricuspid or pulmonary valve regurgitation or evident signs of pulmonary hypertension (PH) were observed [9,13]; however, the pulmonary-to-systemic blood flow ratio calculated using a Doppler was 2.96. Given the enlargement of the left side of the heart, surgical intervention was deemed necessary, and the cat underwent surgical correction using cardiopulmonary bypass.

### 2.2. Treatment

#### 2.2.1. Anesthesia

Pre-anesthesia included subcutaneous administration of atropine (0.025 mg/kg; Atropine Sulfate; NIPRO ES PHARMA Co., Ltd., Osaka, Japan), followed by intravenous administration of cefazolin (20 mg/kg, with additional doses every 2 h intraoperatively; Cefazolin Sodium for Injection, Nichi-Iko Pharmaceutical Co., Ltd., Toyama, Japan), and slow intravenous injection of dexamethasone (0.2 mg/kg; dexamethasone injection A; Nippon Zenyaku Kogyo Co., Ltd., Fukushima, Japan). The cat was oxygenated with 100% oxygen for 5 min and received an intravenous injection of fentanyl (2 μg/kg; Fentanyl injection 0.5 mg; Janssen Pharmaceutical K.K., Tokyo, Japan). Subsequently, alfaxalone (5 mg/kg; Alfaxan multidose; Meiji Animal Health Co., Ltd., Kumamoto, Japan) was administered intravenously, and anesthesia was maintained with isoflurane (1–2%; Isoflurane; Mylan Pharma Co., Ltd., Osaka, Japan). Intraoperatively, a continuous rate infusion (CRI) of fentanyl (5 μg/kg/h) was administered to manage pain. Rocuronium bromide (0.3 mg/kg; Eslax intravenous; MSD K.K., Tokyo, Japan) was administered intravenously to halt respiration, with additional doses of 0.1 mg/kg administered every 40 min as required. If the blood pressure dropped intraoperatively, isoflurane was discontinued, and anesthesia was maintained with a CRI of alfaxalone (5–8 mg/kg/h).

#### 2.2.2. Surgical Technique

The surgical technique was performed following the procedure outlined in Figure 5.

Cannulation

After anesthesia, the cat was positioned in dorsal recumbency. Arterial pressure was measured invasively after the femoral artery and vein were exposed via cutdown. A median sternotomy was performed, followed by a median pericardiotomy, and the pericardium was retracted to either side to create a pericardial tent. At this stage, palpation of the thrill due to the VSD revealed a defect in the right ventricular outflow tract, and the VSD was approached by incising the pulmonary artery.

The periaortic adipose tissue was removed to expose the aorta for the placement of a perfusion cannula distally and a myocardial protection solution root cannula proximally. Before inserting the perfusion cannula, double purse-string sutures were placed using non-absorbable monofilament 6-0 polyvinylidene fluoride sutures (Asflex; Konoseisakusyo Co., Ltd., Tokyo, Japan). Similarly, a single purse-string suture was used to secure the root cannula. For venous drainage, the cranial vena cava, azygos vein, and caudal vena cava were dissected, and taping was performed on the cranial and caudal vena cava using an expanded polytetrafluoroethylene suture (CV-0; ethylene oxide gas before surgery). A purse-string suture with 6-0 polyvinylidene fluoride was placed before the venous drainage cannula was inserted. A purse-string suture of the same material was used to vent the left atrial appendage. Carbon dioxide was infused into the thoracic cavity during cardiac incision to prevent air embolism.

Subsequently, heparin (heparin sodium injection; NIPRO ES PHARMA Co., Ltd., Osaka, Japan) was administered intravenously at a 200 IU/kg dose, and an activated coagulation time (ACT) exceeding 300 s was confirmed. An 8 Fr arterial perfusion cannula (DLP™ One-Piece Pediatric Arterial Cannula, MEDTRONIC JAPAN CO., LTD., Tokyo, Japan) and a 16 G root cannula (DLP™ Pediatric Aortic Root Cannulae, MEDTRONIC JAPAN CO., LTD., Tokyo, Japan) were then inserted and secured with tourniquets. An 8 Fr curved venous return cannula was inserted into the cranial vena cava, and a 10 Fr straight venous return cannula (Flexmate, Senko Medical Instrument MFG. Co. Ltd., Tokyo, Japan) was placed in the caudal vena cava, inserted distal to the taped ePTFE sutures, and secured with a tourniquet. A vent cannula was placed in the left atrium and secured using a tourniquet.

Partial bypass perfusion was initiated, followed by cross-clamping of the aorta. Myocardial protection solution (Miotector coronary vascular injection, Kyowa Criticare Co., Ltd., Atsugi City, Kanagawa, Japan) was slowly administered through a root cannula at 34 mL/kg (150 mL) to achieve cardiac arrest and transition to total bypass perfusion. After 30 min, an additional dose of 15 mL/kg (65 mL) was administered, totaling 215 mL. Initially, the plan was to tape the azygos vein together with the cranial vena cava. However, owing to the distance, a suction cannula was inserted through the right atrial incision to perform suction venous drainage from the blood perfused from the azygos vein.

2.VSD Closure

After cardiac arrest was induced, the main pulmonary artery was incised, and the pulmonary valve was incised to the right ventricle at the commissure to expose the ventricular septum. This revealed an elliptical defect surrounded by muscular tissue, measuring 9 mm in length and approximately 2–3 mm in width (Figure 6A). The full extent of the defect was visualized by inserting a right-angle clamp into the defect and pulling it toward the operator. The rim of the defect was sutured using six double-armed sutures with polypropylene pledgets (Oval-M; Matsudaika Inc., Tokyo, Japan). A patch (Nippon Becton Dickinson Company, Ltd., Tokyo, Japan) was used to close the defect, which was tailored to fit the size of the hole. Double-armed needles previously sewn into the rim were then threaded through the circular patch in sequence, and closure was completed once it was confirmed that no blood had leaked from the left ventricle (Figure 6B).

3.Weaning and Chest Closing

Blood was evacuated from the left atrial vent cannula, and the pulmonary artery and right ventricle were continuously sutured using 6-0 polyvinylidene fluoride sutures. The right atrium was similarly sutured continuously, and upon release of the aortic cross-clamp, the spontaneous beating of the heart resumed. Partial cardiopulmonary bypass continued until the patient was rewarmed to 36.5 °C. The total aortic cross-clamp time was 56 min, and the cardiopulmonary bypass duration was 105 min. Subsequently, a chest drain was placed, and the chest was closed using the standard method. Postoperative pain was managed by administering fentanyl (2 µg/kg/h) continuously for 24 h.

### 2.3. Complications

The packed cell volume (PCV) decreased to 15.7% postoperatively, necessitating a transfusion of 49 mL of blood, after which the PCV on the following day was 24.3%. Urine output immediately post-surgery was maintained at 2 mL/kg/h but subsequently decreased, and by the second postoperative day, azotemia was observed (blood urea nitrogen [BUN]: 78.5 mg/dL; creatinine [Cre]: 3.6 mg/dL). The blood potassium level remained normal at 4.03 mmol/L. Continuous fluid therapy resulted in the normalization of values by postoperative day 12 (BUN: 26.0 mg/dL; Cre: 1.2 mg/dL). However, by postoperative day 7, the pleural effusion had turned milky, requiring approximately 100 mL of it to be drained daily. The triglyceride (TG) and total cholesterol (T-Chol) levels in the pleural fluid were 60.0 mg/dL, with serum TG and T-Chol levels at 59.0 mg/dL and 183.0 mg/dL, respectively. From postoperative day 8, oral administration of rutin 500 mg thrice daily was initiated. A contrast-enhanced computed tomography scan under anesthesia was performed on postoperative day 12. The thoracic duct was highlighted using an iodinated contrast (iohexol 300 mg/mL; Fuji Pharma Co., Ltd., Tokyo, Japan) injected subcutaneously at 1.8 mL/kg around the anus, followed by 5 min of massage. This revealed significant collateral development in the anterior chest and leakage of the contrast agent into the pleural cavity. No thrombi or embolic material was found in the cranial vena cava. The milky effusion gradually decreased and resolved by postoperative day 15. This allowed for the removal of the chest drain the following day and discharge on postoperative day 17. Antithrombotic therapy was not implemented in this case.

### 2.4. Long-Term Follow-Up

On postoperative day 490, the patient remained asymptomatic and lively, with no recurrence of the chylothorax. Echocardiography showed slight residual shunting from the VSD (Figure 7A), but the pulmonary artery had returned to almost normal (MPA: Ao 0.93; 8.3/8.9 mm), and LA/Ao and LVIDd were within normal ranges at 1.41 (1.25/0.89 cm) and 1.29 cm, respectively (Figure 7B). IVSd and LVFWd measured 0.54 cm and 0.50 cm, respectively; IVSd was mildly thick (IVSd: 0.39 cm, range: 0.29–0.51 cm; LVWDd: 0.38 cm, range: 0.29–0.50 cm [12]). IVSs and LVFWs were 0.63 cm and 0.72 cm, respectively; they were within the normal range (IVSs: median 0.62 cm, IQR: 0.44–0.87 cm; LVFWs: 0.65 cm, IQR: 0.48–0.87 cm [12]). LVIDs was 0.68 cm in the normal range (median 0.86 cm, IQR: 0.57–1.30 cm), and FS% was 47.22% (normal range: median 45%, IQR: 28–62% [12]). The patient is undergoing annual check-ups for suspected hypertrophic cardiomyopathy.

## 3. Discussion

We encountered a cat with an infundibular muscular VSD classified according to the Soto classification. We surgically closed the defect using total bypass perfusion through a median sternotomy approach with a heart-lung machine. To our knowledge, there are no prior reports of the surgical closure of VSD in cats, and none specifically detailing infundibular muscular VSD. Additionally, while cardiac surgeries in cats using cardiopulmonary bypass, including left atrial-to-atrial septal defects, have been reported, these were conducted through an intercostal approach and involved partial bypass perfusion without cardiac arrest induced by cardioplegia [14,15].

Initially, the cat was diagnosed with the most common type of VSD in felines, perimembranous VSD, through echocardiographic examination. However, upon opening the chest and palpating the heart, the thrill observed in the pulmonary outflow tract led to a diagnosis of infundibular muscular VSD. In felines, VSDs are generally classified according to the Kirklin classification, where Type 1 VSDs are supracristal, Type 2 are membranous or perimembranous, and Types 3 and 4 are atrioventricular canal and muscular VSDs, respectively, with 79% diagnosed as membranous or perimembranous [3].

However, in this case, the VSD was located in the outflow septum and surrounded by muscular tissue, making it difficult to classify using the Kirklin classification. The Soto classification, commonly used in humans, categorizes VSDs into four types: (1) infundibular defects situated in the infundibular septum, often part of which are the semilunar valves; (2) membranous defects centered around the membranous septum and extending to areas near the atrioventricular or aortic valves; (3) defects between the membranous part and the inflow tract, similar to those observed in atrioventricular septal defects; and (4) muscular defects, where the surrounding tissue of the defect is entirely muscular [7]. Furthermore, muscular defects are sub-classified based on their location: infundibular, inflow tract, and trabecular septums.

According to the Kirklin classification, a Type 1 defect located above the crista supraventricularis corresponds to an infundibular defect in the Soto classification, where the aortic valve may deviate. In the Kirklin classification, Type 1 defects are located above the crista supraventricularis and correspond to infundibular defects in the Soto classification. These defects are large and occur directly below the pulmonary valve, potentially causing deviations in the aortic valve. However, in the present case, although the defect was near the crista supraventricularis, it was not an infundibular defect due to its distance from the pulmonary artery. Consequently, it was ultimately classified as a muscular defect according to the Soto classification and diagnosed as an infundibular muscular VSD. This type of defect is advantageous for suturing because, unlike perimembranous VSDs, it does not have a nearby conduction system.

In the case of membranous defects, color Doppler imaging using the clock-face method typically reveals a defect between the 10 and 11 o’clock positions. Conversely, a defect under the aortic valve (subaortic VSD) appears between the 11 and 1 o’clock positions, and in the Kirklin classification, a Type 1 defect is observed between the 1 and 2 o’clock positions [16]. In the present case, the absence of a defect directly beneath the pulmonary artery and the shunt flow originating from the 12 o’clock direction suggested that it was unlikely to be an infundibular or membranous defect. Therefore, a preoperative diagnosis of infundibular muscular VSD may have been possible.

Additionally, there was a slight discrepancy between the measurements of VSD size by using color Doppler imaging and those obtained visually intraoperatively. This discrepancy could be attributed to the difference between the beating heart and the heart relaxed by cardioplegia and the elliptical shape of the defect. Studies on humans have reported that three-dimensional (3D) echocardiography can measure the size and location of defects more accurately than that measured using two-dimensional echocardiography [17].

Henceforth, it would be beneficial to use preoperative 3D echocardiography to better understand the size and location of defects, particularly in cats undergoing closure procedures. Additionally, the Soto classification appears to be more appropriate for categorizing VSDs, especially when traditional methods such as the Kirklin classification encounter limitations due to anatomical variations specific to felines.

Shunt volume in VSD is determined by the size of the defect, pulmonary vascular resistance, and systemic vascular resistance, with severity typically assessed by the Qp:Qs ratio. When Qp:Qs is <1.5, between 1.5 and 2.5, and >2.5, the VSD is defined as small (restrictive), moderate (moderately restrictive), or large (nonrestrictive), respectively, correlating with the VSD: Ao ratio [3]. A higher Qp:Qs ratio is associated with pulmonary hypertension (PH) and heart failure; in this case, both high Qp:Qs and a high VSD: Ao ratio were observed, along with significant dilation of the main pulmonary artery. When Qp:Qs is <1.5, it is advisable to monitor the patient’s condition without any intervention. However, if Qp:Qs exceeds 2.5 and there are clinical signs such as cardiac enlargement or indicators suggestive of PH, aggressive therapeutic intervention is warranted [3].

Surgical correction using a cardiopulmonary bypass is less common in cats than in dogs. In dogs, it is common to establish a cardiopulmonary bypass by inserting a perfusion cannula into the common carotid artery; however, in cats, this artery is too narrow to accommodate the necessary equipment. Additionally, cats have a smaller blood volume, which complicates securing sufficient blood for transfusion and leads to anemia due to blood dilution during cardiopulmonary bypass.

In cats, surgical corrections for conditions such as Cor Triatriatum Sinister (CTS) and atrial septal defect (ASD) have been reported, utilizing intercostal approaches without the use of cardioplegia, thereby avoiding blood dilution [14,15]. In ASD repair, a 5 Fr feeding tube is used as a substitute for a perfusion cannula and inserted into the common carotid artery [15]. For CTS, an 8 Fr perfusion cannula was inserted into the descending aorta [14]. In our case, because a median sternotomy approach was used, we were able to insert an appropriately sized perfusion cannula into the ascending aorta without difficulty, thus maintaining cerebral circulation due to the direction of blood flow. For venous drainage, the ASD cases involved the insertion of 8 Fr straight-type cannulas into the cranial and caudal vena cava [15]. However, in CTS, a 16 Fr cannula was inserted into the right atrium [14]. In this case, we chose a curved venous return cannula for the cranial vena cava to facilitate insertion at the tip of the cannula.

Unlike the previous two cases, the use of cardioplegia in this case provided a stable field for surgery, making the procedure easier. However, 215 mL of cardioplegia was administered, which resulted in postoperative anemia. The transient renal impairment observed in this case may have been associated with anemia [18]. Given that hypothermia during cardiopulmonary bypass can reduce the heart rate, surgery under pulsatile conditions might have been possible [15,19]. Furthermore, although not explored in cats, techniques such as electrically induced ventricular fibrillation have been reported in dogs [20]. In cats, where the donor blood supply is often insufficient, future considerations may need to include using less cardioplegia or avoiding it altogether.

In veterinary medicine, open-heart surgery typically involves an intercostal thoracotomy approach. The right intercostal approach is appropriate for treating common membranous VSDs in cats. This allows the insertion of venous drainage cannulas into the cranial and caudal vena cava, enabling VSD closure through an incision in the right atrium or ventricle. However, in this case, an infundibular muscular VSD was diagnosed after thoracotomy at the left ventricular outflow tract. Such a diagnosis would likely have resulted in an unsuccessful outcome with the right intercostal approach, justifying the use of a median sternotomy approach to prepare for unforeseen circumstances.

Additionally, the patient exhibited transient chylothorax postoperatively, which resolved with conservative treatment. Chylothorax is a rare complication in humans following thoracotomy, primarily caused by thoracic duct damage, with no significant difference in incidence between the intercostal and median sternotomy approaches [21]. In cats, the thoracic duct runs along the left side of the aorta and drains into the venous system [22]. The possibility of thoracic duct damage cannot be excluded in this case because the ascending aorta was dissected to insert perfusion and root cannulas using the median sternotomy approach. Therefore, careful handling of major vessels is necessary when performing a median sternotomy.

Postoperative chylothorax can also be associated with elevated central venous pressure during surgery [23]. In this case, the central venous pressure might have increased due to taping of the cranial vena cava, potentially contributing to thoracic duct damage. However, similar to other cases in cats in which chylothorax does not necessitate surgery unless the thoracic duct is severed, this condition improved with the administration of rutin and regular drainage from a chest drain [24]. Therefore, conservative treatment and observation over 1–2 weeks are suitable for managing postoperative chylothorax following open-heart surgery.

## 4. Conclusions

This is the first reported instance of encountering an infundibular muscular VSD in a cat and closing it surgically using the cardiopulmonary bypass approach. Using cardioplegia to achieve cardiac arrest during cardiopulmonary bypass in cats facilitates surgical manipulation; however, it may lead to postoperative anemia due to blood dilution. Transient chylothorax can also occur. Improvements have been made to this approach; however, median sternotomy for open heart surgery in cats appears to be a viable option for treating congenital heart disease.

## Figures and Tables

**Figure 1 animals-14-01736-f001:**
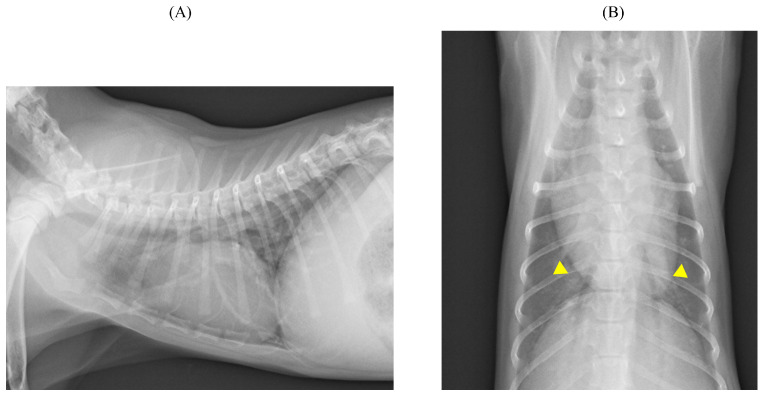
Thoracic radiographic findings. (**A**) Lateral view showing a vertebral heart size of 9.0 vertebrae, which is within the normal range for kittens. (**B**) The dorsoventral view showing a cardiothoracic ratio of 74.8%, which is within the normal range for kittens, although an enlarged pulmonary artery shadow is observed (arrowhead).

**Figure 2 animals-14-01736-f002:**
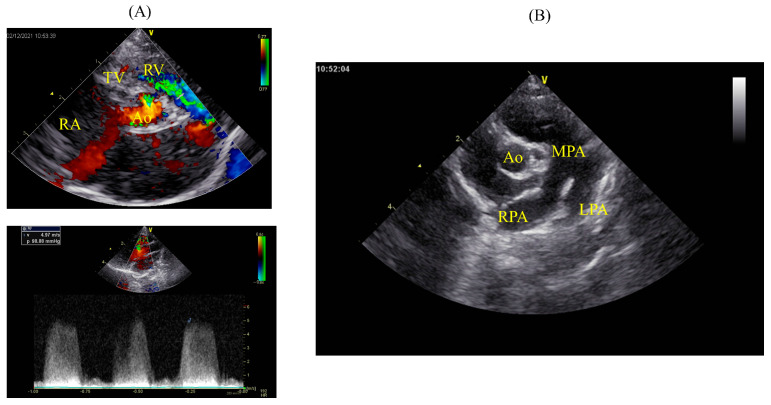
Echocardiographic findings on Day 1. ((**A**), upper) Right parasternal short−axis view at the level of the aortic valve using a color flow Doppler. The left−to−right shunt from the VSD is located between the pulmonary artery outflow tract and tricuspid valve, consistent with a combination of Type 2 and 3 VSD according to the Kirklin classification. ((**A**), lower) Using the right parasternal left ventricular outflow tract view, the left−to−right shunt from the VSD was measured with continuous wave Doppler and recorded as 4.97 m/s. (**B**) Right parasternal short−axis view at the level of the aortic valve showing marked dilation of the pulmonary artery (MPA/Ao: 1.48). Ao: aorta; RA: right atrium; TV: tricuspid valve; RV: right ventricle; MPA: main pulmonary artery; RPA: right pulmonary artery; LPA: left pulmonary artery.

**Figure 3 animals-14-01736-f003:**
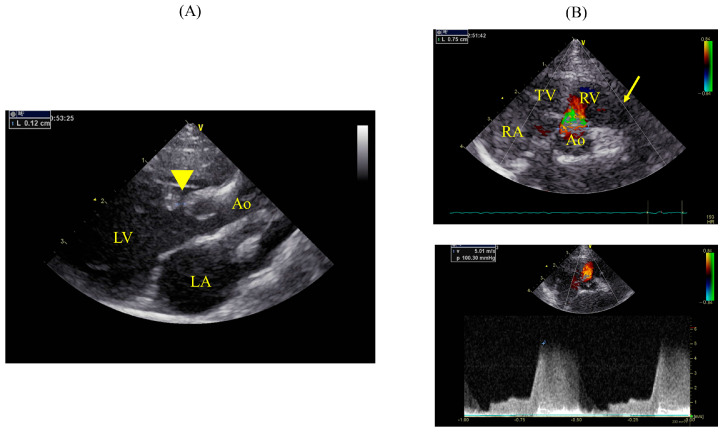
Echocardiographic findings on day 383. (**A**) The right parasternal long-axis view of the LV outflow tract showing a VSD directly below the aortic valve (yellow arrowhead). ((**B**), upper) The right parasternal short-axis view at the level of the aortic valve using a color flow Doppler, where the largest diameter of the VSD is observed during diastole. The left−to−right shunt from the VSD is in the 12 o’clock direction, located between the TV and pulmonary valve (yellow arrow). ((**B**), lower) The left−to−right shunt from the VSD was measured with continuous wave Doppler and recorded as 5.01 m/s. LA: left atrium; LV: left ventricle; Ao: aorta; RA: right atrium; TV: tricuspid valve; RV: right ventricle; VSD, ventricular septal defect.

**Figure 4 animals-14-01736-f004:**
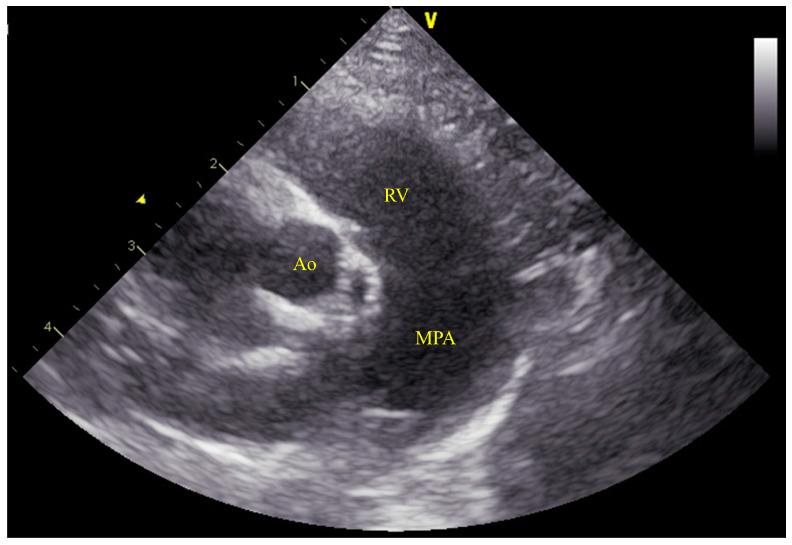
Right parasternal short-axis view at the level of the aortic valve showing significant dilation of the MPA. The aorta and MPA diameters are 0.85 cm and MPA 1.10 cm, respectively Ao ratio of 1.29. Ao: aorta; MPA: main pulmonary artery; RV: right ventricle.

**Figure 5 animals-14-01736-f005:**
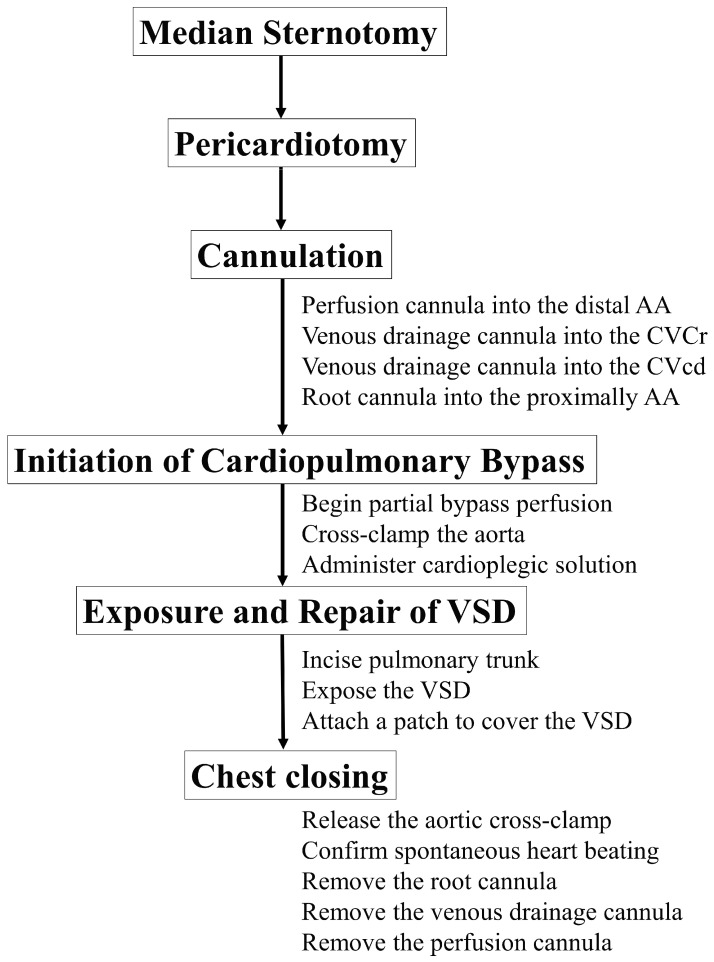
Procedure diagram for ventricular septal defect closure in the cat. AA: Ascending aorta; CVcr: Cranial vena cava; CVCd: Caudal vena cava; VSD: ventricular septal defect.

**Figure 6 animals-14-01736-f006:**
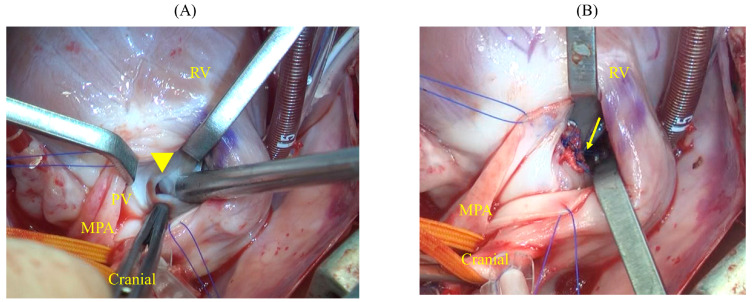
Intraoperative findings of the VSD. (**A**) The MPA is incised, and an incision from the commissure of the PV to the RV reveals a VSD measuring 9 mm in length and 2–3 mm in width (yellow arrowhead). The defect is observed on the RV side of the PV. (**B**) The VSD is closed by using a patch (yellow arrow). MPA: main pulmonary artery; PV: pulmonary valve; RV: right ventricle; VSD: ventricular septal defect.

**Figure 7 animals-14-01736-f007:**
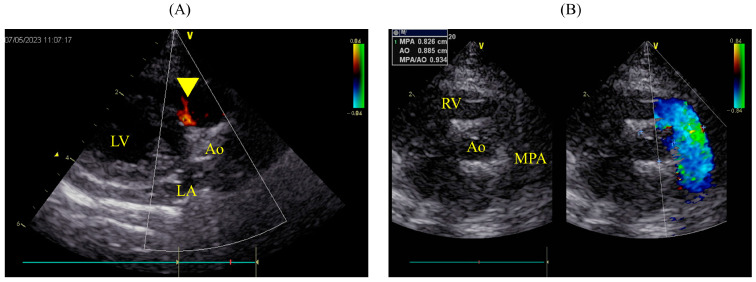
Echocardiographic findings 490 days postoperatively. (**A**) Right parasternal long−axis view of the LV outflow tract, where a color flow Doppler is slightly adjusted to confirm residual shunting, showing minor residual shunting (yellow arrowhead). (**B**) Simultaneous right parasternal short-axis view at the level of the aortic valve indicating the diameters of the aorta and MPA at 8.9 mm and 8.3 mm, respectively, with an MPA: Ao ratio of 0.93. LA: left atrium; LV: left ventricle; Ao: aorta; MPA: main pulmonary artery; RV: right ventricle.

## Data Availability

The original contributions presented in the study are included in the article. Further inquiries can be directed to the corresponding author.

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
