# Peer review of "Surgical Correction of Infundibular Muscular Ventricular Septal Defect in a Cat: A Case Report"

_animals, 2024, doi:10.3390/ani14121736_

Round 1

Reviewer 1 Report

Comments and Suggestions for Authors

Dear Authors,

I read your case report with a great interest. In my opinion, it is a valuable paper in the veterinary medicine, widening our knowledge of the possibilities in small animal surgery.

Some minor comments are presented below:

- lines 23-24: please revise the sentence

- lines 70-71: it is unclear how cardiac enlargement was noted during vaccination? was the radiographic examination taken? please clarify

- lines 81-90: please provide the images from the initial TTE

- lines 93 and further: what was the clinical condition of the cat in day 383? Did it show any signs of heart disease?

- line 108: please specify why a normal value for dogs (and not cats) is presented

Author Response

Response to reviewer 1

Manuscript ID: animals-3028038

Title: Surgical Correction of Infundibular Muscular Ventricular Septal Defect in a Cat: A Case Report

Thank you for your valuable comments and opinions. Please find below our point-by-point responses to your comments. Corrections to the text of the paper are indicated with yellow highlights.

- lines 23-24: please revise the sentence

Response: Thank you for pointing this out. We have corrected the sentences you pointed out and asked the English proofreading company (www.editage.com) to proofread them again.

- lines 70-71: it is unclear how cardiac enlargement was noted during vaccination? was the radiographic examination taken? please clarify

Response: Thank you for your suggestion. At the animal hospital that referred the case to our clinic, a thoracic radiography was taken before a vaccination, and an enlarged heart was subjectively determined. This information has been added.

- lines 81-90: please provide the images from the initial TTE

Response: Thank you for your suggestion. We have added the initial echocardiography results.

- lines 93 and further: what was the clinical condition of the cat in day 383? Did it show any signs of heart disease?

Response: Thank you for pointing this out; we have added it, as there were no obvious clinical signs of suspected cardiogenicity at 383 days.

- line 108: please specify why a normal value for dogs (and not cats) is presente

Response: Thank you for pointing this out. We have corrected it to the reference values in cats and changed the references.

Reviewer 2 Report

Comments and Suggestions for Authors

The authors have prepared a pragmatic description of the diagnostic and treatment management of the Ragdoll cat. The heart defect described in the above article is a new case in the literature.

Introduction

Lines

49 - It is also worth mentioning that cardiac patients have heart defects in about 10% of cases.

51 - Please also provide other citation and data on the percentage that VSD represents. E.g. in another paper the frequency is 18%. This is important information as the frequency of feline heart defects show regionalisation.

(Schrope, D.P. Prevalence of Congenital Heart Disease in 76,301 Mixed-Breed Dogs and 57,025 Mixed-Breed Cats. J Vet Cardiol 2015, 17, 192-202, doi:10.1016/j.jvc.2015.06.001.)

Structure of the case report

The entire case report is a single text. This is not practical and makes it difficult to orient yourself when you want to return to some information. Please divide the whole text.

Please add a subsection: 

Surgical technique. Include the surgical technique. Enrich the whole subsection with a procedure diagram.

Patient. Describe the patient and why he/she was referred for consultation. That is, everything about the Ragdoll cat in the initial "Case description" section.

Treatment. Here describe the course of the operation from anaesthesia to awakening (anaesthesia to be listed as a sub-item). That is, the exact course including access modifications (cannulation through the right atrium), complications (thoracic duct), etc.

Another sub-point to add is OUTCOME AND FOLLOW-UP. Provide the level of flow wave reduction and other postoperative information. Please also apply the notes on images here.

Presentation of the case report

The case report must present accurate diagnostic documentation. Imaging techniques play a major role in cardiology, so please prepare a better quality presentation of the results.

Please prepare videos in the supplements showing the flow through the VSD before and after surgery.

Please prepare a video of several minutes of the operation. One photo of the operation does not tell me anything. 

X-ray scans with enlarged heart and calculated VHR are missing.

M-mode presentation of heart wall thickness is missing.

101 - the leakage waveform spectrum where you can see the flow velocity is missing.

220 - this projection is of too low quality. If it is only possible to present residual flow in this projection, please additionally provide images from the classic projection. For example in the supplement as a video.

Best regards

Author Response

Response to reviewer 2

Manuscript ID: animals-3028038

Title: Surgical Correction of Infundibular Muscular Ventricular Septal Defect in a Cat: A Case Report

49 - It is also worth mentioning that cardiac patients have heart defects in about 10% of cases.

Response: Thank you for pointing this out. You are correct. We have added your remarks to the text.

51 - Please also provide other citation and data on the percentage that VSD represents. E.g. in another paper the frequency is 18%. This is important information as the frequency of feline heart defects show regionalisation.

(Schrope, D.P. Prevalence of Congenital Heart Disease in 76,301 Mixed-Breed Dogs and 57,025 Mixed-Breed Cats. J Vet Cardiol 2015, 17, 192-202, doi:10.1016/j.jvc.2015.06.001.)

Response: Thank you for pointing this out. You are correct. We have added your remarks to the text.

Structure of the case report

The entire case report is a single text. This is not practical and makes it difficult to orient yourself when you want to return to some information. Please divide the whole text.

Please add a subsection:

Surgical technique. Include the surgical technique. Enrich the whole subsection with a procedure diagram.

Patient. Describe the patient and why he/she was referred for consultation. That is, everything about the Ragdoll cat in the initial "Case description" section.

Treatment. Here describe the course of the operation from anaesthesia to awakening (anaesthesia to be listed as a sub-item). That is, the exact course including access modifications (cannulation through the right atrium), complications (thoracic duct), etc.

Another sub-point to add is OUTCOME AND FOLLOW-UP. Provide the level of flow wave reduction and other postoperative information. Please also apply the notes on images here.

Response: Thank you for pointing this out. As you indicated, we have split the "Case Description" and created a subsection. We have also created a diagram showing the surgical procedure.

Presentation of the case report

The case report must present accurate diagnostic documentation. Imaging techniques play a major role in cardiology, so please prepare a better quality presentation of the results.

Please prepare videos in the supplements showing the flow through the VSD before and after surgery.

Please prepare a video of several minutes of the operation. One photo of the operation does not tell me anything.

Response: Thank you for pointing this out. We will upload a video of the surgery and a video showing the shunt's flow from the VSD pre- and postoperatively as supplemental data.

X-ray scans with enlarged heart and calculated VHR are missing.

Response: Thank you for pointing this out. We have added Thoracic cardiography to the text.

M-mode presentation of heart wall thickness is missing.

Response: Thank you for pointing this out. We have added other M-mode data to the text.

101 - the leakage waveform spectrum where you can see the flow velocity is missing.

Response: Thank you for pointing this out. We have added shunt's flow from the VSD to the diagram.

220 - this projection is of too low quality. If it is only possible to present residual flow in this projection, please additionally provide images from the classic projection. For example in the supplement as a video.

Response: Thank you for drawing our attention to this. We will upload a video of the surgery and a video showing the shunt's flow from the VSD pre- and postoperatively as supplemental data.